# The Differences in Protein Degradation and Sensitization Reduction of Mangoes between Juices and Pieces Fermentation

**DOI:** 10.3390/foods12183465

**Published:** 2023-09-17

**Authors:** Mengtian Tian, Qiuqin Zhang, Xianming Zeng, Xin Rui, Mei Jiang, Xiaohong Chen

**Affiliations:** 1Sanya Institute of Nanjing Agricultural University, Nanjing Agriculture University, Sanya 572024, China; 2022808176@stu.njau.edu.cn (M.T.); zxm@njau.edu.cn (X.Z.); ruix@njau.edu.cn (X.R.); meijiang9@njau.edu.cn (M.J.); xhchen@njau.edu.cn (X.C.); 2College of Food Science and Technology, Nanjing Agriculture University, Nanjing 210095, China

**Keywords:** mango, allergy, fermentation, protein, kombucha

## Abstract

Given the allergic reaction caused by mangoes, nonthermal food technologies for allergenicity reduction are urgently desired. This study aimed to assess the impact of kombucha fermentation on the allergenicity of mangoes. The total proteins, soluble proteins, peptides, amino acid nitrogen, the SDS–PAGE profiles of the protein extracts, and immunoreactivity of the sediment and supernatant were measured in two fermentation systems (juices and pieces fermentation). Throughout the fermentation, the pH decreased from about 4.6 to about 3.6, and the dissolved oxygen reduced about 50% on average. However, the protein degradation and sensitization reduction of mangoes were different between the two fermentation systems. In juices fermentation, there was a drop in proteins and peptides but an increase in amino acids, due to the conversion of proteins and peptides into amino acids both in the supernatant and sediment. The allergenicity decreased both in the solid and liquid phases of juices fermentation. In pieces fermentation, proteins and peptides were decreased in the solid phase but increased in the liquid phase. This was due to the fact that proteins and peptides were partly transported into the culture liquid, resulting in a decrease of allergenicity in fruit pieces and an increase in culture liquid. The principal component analysis results showed that the fermentation type had significant effects on the protein degradation and sensitization reduction, while mango variety had no significant effect. These results demonstrate that kombucha fermentation can reduce the allergenicity of mangoes, and it is more effective in juices fermentation than in pieces fermentation. The present study provides a theoretical basis for developing hypoallergenic mango products.

## 1. Introduction

Mango (*Mangifera indica* L.), considered as one of the most consumed fruits, is grown in many parts of the world, especially in tropical regions. This fruit is cultivated in 85 countries worldwide, with a cultivated area of approximately 5 million ha [1]. Mangoes accounted for more than half of the global production of the major tropical fruits [2]. The Food and Agriculture Organization of the United Nations (FAO) reports mango, mangosteen, and guava production as one aggregated category, the global production volume of which reached 57 million metric tons in 2021 [3]. Mango fruit contains a number of antioxidant substances, such as vitamins, dietary fibers, polyphenols, and fatty acids [4]. As a nutrient-rich fruit, mango supplementation is associated with nutrient intake, diet quality, and health outcomes [5]. Mango consumption has been proven to be beneficial for reducing metabolic diseases and improving the gut microbiome [6,7]. Hence, mango products, including juice, powder, extract suppliers, etc., are popular among customers.

Mangoes can cause an allergic reaction, with a common clinical manifestation of contact dermatitis [8]. There is a cross-reaction between mango fruit and other foods, resulting in an increase in the frequency of allergic symptoms [9,10]. The major IgE-related allergens in mango products are identified as proteins, such as Man i 1 (glyceraldehyde 3-phosphate dehydrogenase), Man i 2 (type I chitinases), Man i 3 (profilin), β-1,3-glucanase, lipoxygenase, etc. [10,11,12,13]. In recent years, with the increase in the number of people suffering from fruit allergies, the demand for hypoallergenic foods has been increasing worldwide [14,15]. Therefore, food processing technologies that can reduce the allergenicity of mango products are urgently needed.

Selecting a safe and efficient protocol is the key to developing hypoallergenic food products. Reducing food allergenicity can be achieved by various strategies, including thermal treatment, irradiation, ultrasonication, polyphenol supplement, enzymatic crosslinking, fermentation, etc. [14,16,17,18,19,20,21]. In industrial food manufacture, thermal processing can destroy heat-sensitive nutrients and food product qualities such as flavor, color, and texture. Mangoes will brown and produce off-odors after thermal treatment [22]. Moreover, it was found that the mango allergens were very stable during enzymatic matrix decomposition and heating [23]. Fermentation is used as a nonthermal processing technology to reduce allergenicity and improve the taste, flavor, digestibility, and texture of food products [24]. Physical processes reduce allergenicity by altering the secondary and tertiary structures of proteins related to allergenicity, while biochemical processes cleave the allergenic protein epitopes [17]. Protein may restore its allergenicity after physical treatment, but it can completely lose its allergenicity after biochemical treatment [16]. The common biochemical approaches for reducing or eliminating food allergenicity are fermentation and enzymatic hydrolysis. Compared with the enzymatic method, fermentation has unique advantages in reducing the allergenicity of food due to its possibility to degrade multiple allergenic proteins and improve the nutritional value and physicochemical properties of food materials [21,24,25]. The microorganisms most commonly involved in allergenicity reduction are lactic acid bacteria (LAB) [25,26]. LAB have been widely applicated in the fermentation of wheat, dairy, and legumes products [27,28,29,30]. The sensory acceptability, including aroma, natural taste, sweet, and flavor, of mango juices may be reduced by a single strain of LAB fermentation [31]. Co-fermentation with different strains of microorganisms is more conducive to improving the quality of mango products [32]. Kombucha, a symbiotic culture of yeast and bacteria, especially acetic acid bacteria and LAB, can be used to ferment various fruits, such as pomegranate, red grape, sour cherry, and apple [33]. Previously, we found that mango juices fermented by kombucha were accepted with a good flavor. In this study, kombucha was used in mango fermentation instead of LAB. Hence, kombucha fermentation is considered as a natural and potential nonthermal allergy-reducing technique with a positive perception by consumers.

Allergic reactions to mango fruit have become increasingly important; nevertheless, few reports of strategies for reducing mango allergens have been mentioned in the literature. The aim of this study was to explore the possibility of kombucha fermentation for mangoes after beating for their juices or cutting them into pieces. The microbial activity was monitored by measuring the changes of pH and DO during fermentation. In order to investigate the changes in proteins and sensitizations, the total proteins, soluble proteins, peptides, amino acids, the SDS–PAGE profiles of the protein extracts, and IgE immunoreactivity of the sediment and supernatant were measured in two fermentation systems (juices and pieces fermentation). By comparing the differences between the two fermentation systems (juices and pieces fermentation), the study aims to provide a theoretical basis for developing hypoallergenic mango products.

## 2. Materials and Methods

### 2.1. Materials

Three varieties of mangoes, including Tainung (T-Mango), Keitt (K-Mango), and Green Kaew Mango (Q-Mango), were bought from local markets. Kombucha was purchased from Jining Shanghaojiapin Commercial and Trading Co., Jining, China.

Kombucha was cultured in a tea syrup culture medium (containing 4 g/L tea water and 100 g/L sucrose) at 30 °C for 10 days. After incubation, kombucha culture fluid was extracted using a sterile cylinder (ML-32269, Fangge Commercial and Trading Co., Guangzhou, China).

The serum from five allergic sufferers were bought from Chongqing Wolcavi Biological Technology Co., Ltd., (Chongqing, China). The five allergic sufferers were 27, 34, 36, 47, and 30 years old and had specific IgE levels of 16.4, 4.77, 4.24, 4.15, and 6.03 KU L−1, respectively.

Most chemicals and reagents were purchased from Sinopharm Chemical Reagent Co., ltd., (Shanghai, China).

### 2.2. Fermentation of Mango Fruits

Mango fruits were fermented after beating (juices fermentation, Figure 1A) or cutting into pieces (pieces fermentation, Figure 1B). For juices fermentation, mango fruits were peeled, sliced into pieces, mixed with equal weights of purified water, and handled with a grinding mill (RS-FS1401, Royalstar, Anhui, China), with 6% sucrose and 15% kombucha culture solution added aseptically. For pieces fermentation, mango fruits were peeled and sliced into pieces (3 cm × 2 cm × 0.5 cm) and mixed with an equal weight of purified water. The samples were fermented with a glass fermenter (BIOTECH-1BG-3-5BG, Shanghai Baoxing Bio-Engineering Equipment Co., Ltd., Shanghai, China) at 25 °C for 24 h. 

### 2.3. PH and Dissolved Oxygen (DO)

The pH and DO of the fermentation solutions were automatically measured via pH (InPro3100, METTLER TOLEDO, Zurich, Switzerland) and oxygen probes (InPro 6860i, METTLER TOLEDO, Zurich, Switzerland).

### 2.4. Crude Protein Content

The crude protein content was determined by the Kjeldahl method with a Kjeldahl nitrogen determinator (KjeltecTM8400, FOSS, Copenhagen, Danmark) [34]. Six milliliters of the sample were accurately weighed out and then digested with ten millimeters of concentrated sulfuric acid in the presence of a catalyst by using a Digestor (TecatorTM Digestor2520, FOSS, Copenhagen, Danmark). The catalyst was a mixture of cupric selenite (0.2 g) and potassium sulfate (0.3 g). The method used 40% NaOH to produce an alkaline distillation environment and 4% boric acid solution to collect the distilled ammonia. The titrations were performed with standardized 0.1 N hydrochloric acid. Tashiro’s indicator (0.375 g of methyl red and 0.250 g of methylene blue in 300 mL of 95% ethanol) was used to identify the end point of the titration.

### 2.5. Soluble Protein Content

The soluble protein was measured following the Bradford assay [35]. The samples were centrifuged at 8000 r/min for 10 min. Then, 5 μL of supernatant were mixed with 200 μL of Coomassie Blue staining solution and incubated for 5 min at room temperature. The absorptive value was measured at 595 nm. The protein content was calculated based on the standard curve of bovine serum protein.

### 2.6. Peptide Content

The peptide content was measured according to the method described by Rui et al. [36]. The samples were centrifuged at 8000 r/min for 10 min. The supernatant was mixed with an equal volume of 10% trichloroacetic acid (*w*/*w*) for 1 h and then centrifugated for 10 min at 10,000 r/min. Then, 50 μL of supernatant was reacted with a preprepared 2 mL reagent for 2 min, and then, the absorbent value was measured at 340 nm. The reagent (50 mL) was composed of 25 mL of 100 mmol/L borax, 2.5 mL of 20% (*w*/*w*) sodium dodecyl sulfate, 40 mg of o-Phthaldialdehyde solution dissolved in 1 mL of methanol, and 100 μL of β-mercaptoethanol (β-ME). The peptide content was calculated based on the standard curve of the pancreatic casein building.

### 2.7. Amino Acid State Nitrogen

Ten milliliters of the sample were mixed with seventy millimeters of distilled water and titrated with 0.05 mol/L NaOH solution until the pH value of the solution reached 8.2. Then, 10 mL 38% (*v*/*v*) of formaldehyde solution was added and titrated with NaOH solution until the pH reached 9.2. The volume of NaOH standard solution consumed after the addition of formaldehyde was recorded as V1. Eighty milliliters of distilled water were used as the blank control, and the volume of the consumed NaOH solution in the control was recorded as V2. The amount of amino acid state nitrogen was expressed as the difference between V1 and V2 [37].

### 2.8. SDS-PAGE

Sodium dodecyl sulfate-polyacrylamide gel electrophoresis (SDS-PAGE) was carried out as previously described [38]. The sample was mixed with an equal volume of protein loading buffer (P1016, Solarbio, Beijing, China) and centrifuged at 10,000 r/min for 5 min. Then, 10 μL of the supernatant were loaded into polyacrylamide wells. The gel was stacked at 4%, separated at 12%, and run with a Bio-Rad Miniprotein 3 unit (Bio-Rad Laboratories, Inc., Hercules, CA, USA) at 60 and 120 V for stacking and separating the gel, respectively. The gels were stained with 0.1% (*w*/*v*) Coomassie Brilliant Blue R250 staining solution for 2 h and de-stained with decolorization solution overnight. The protein standard with weight sizes ranging from 10 to 250 kDa was used as a marker. The gels were then scanned with Image Scanner III (GE Healthcare Biosciences, Uppsala, Sweden).

### 2.9. IgE Immunoreactivity

The IgE immunochemical reactivity was measured according to the method described by Peñas et al. [39].

Samples were diluted to 5 μg/mL with 100 mM sodium carbonate buffer (pH 9.6). Then, 100 μL of dilution was added to each well in a 96-well polystyrene ELISA plate (Costar 3590, Corning Inc., New York, NY, USA) and incubated at 4 °C for 18–22 h. The plate was washed thrice with phosphate buffer (pH 7.0, containing 0.05% Tween 20). After drying at room temperature, 250 μL of blocking fluid (containing 2.5% bovine serum albumin) was added into each well and incubated at 37 °C for 2 h. Remove the blocking liquid and wash three times with the phosphate buffer. The plate was incubated with 100 μL of human sera (1:20 diluted with PBS containing 1% BSA, *v*/*v*) at 37 °C for 1 h. The plate was washed again and incubated with 100 μL of antihuman IgE-ε-chain-specific peroxidase (1:2000 dilution with PBS, *v*/*v*) per well. The plate was incubated at 37 °C for 1 h. The wells were washed again, and 100 μL of 3,3′,5,5′-tetramethylbenzidine substrate solution (Sigma Inc., St. Louis, MO, USA) was loaded and incubated for 20 min at 37 °C. Then, 2 M sulfuric acid was added to terminate the reaction. The absorbance was recorded at 450 nm with a BioTek μQuant microplate spectrophotometer (BioTek Instruments, Inc., Winooski, VT, USA).

### 2.10. Statistical Analyses

All the experiments were repeated three times, and the data were reported as averages ± standard deviations. The data were analyzed statistically by Multivariant Analysis of Variance using SPSS 19.0 (SPSS Inc., Chicago, IL, USA), and the means were compared by the multiple range test and *t*-test. Principal component analysis (PCA) was conducted by SPSS 19.0 (SPSS Inc., Chicago, IL, USA).

## 3. Results and Discussion

Based on prior experimental research, the product’s flavor is best when fermented for 24 h; thus, we chose that as the time point. Figure 2A,B,E show the changes of the pH value during juices fermentation. In juices fermentation, the pH values of Green Kaew (Q-Mango), Keitt (K-Mango), and Tainung Mango (T-Mango) decreased from 4.48 to 3.64, from 4.61 to 3.74, and from 4.67 to 4.07, respectively. In pieces fermentation, the initial pH values of T-Mango, K-Mango, and Q-Mango were 4.44, 4.48, and 4.70 and reduced by 0.88, 0.78, and 0.76 after 24 h of fermentation. T-Mango had the lowest initial pH value, followed by K-Mango and Q-Mango. In both fermentation systems, the final pH of Q-Mango was higher than that of T-Mango and K-Mango. Figure 2C,D,F show the change in the dissolved oxygen (DO) levels in the fermentations. The DO concentrations of T-Mango, K-Mango, and Q-Mango, respectively, reduced by 51.60, 53.25, and 56.71 in juices fermentation and 68.10, 60.00, and 60.47 in pieces fermentation. The reductions in the OD values of the three varieties of mangoes in juices fermentation were lower than those in pieces fermentation, indicating that the fermentation process was influenced by fruit morphology.

Monitoring of the fermentation process is related to the improvement in the final product quality. The community of kombucha comprise cultured and uncultivable microorganisms with over 200 microbial species [40]. In different microenvironments, the bacterial component of kombucha is relatively stable, while the yeast species composition is significantly variable [41]. Genomics is the best method for exploring the microbe distribution of kombucha, due to its complex and variable microbial diversity [42,43]. However, this method is time-consuming and costly, and it does not meet the requirements of real-time monitoring of the fermentation process. The results showed that the pH and DO values were monitored instead of microbial detection. During fermentation, the pH and DO values gradually decreased. The reduction in pH was caused by the production of organic acids by the acetic and lactic acid bacteria in kombucha [44,45]. Kombucha cultures can work both in aerobic and anaerobic conditions. In the present study, fermentation was performed in a closed bottle. The consumption of the dissolved oxygen was mainly related to the growth of the producing cellulose bacteria and yeast [46]. This study confirmed that the changes in pH and DO were related to microbial growth, suggesting that these two indicators can be applied in the real-time monitoring of mango fermentation. 

The distribution of the total protein in the liquid and solid phases of the fermentation system was determined (Figure 3). After juices fermentation, the total protein of all sediments and supernatants of the three mangoes, except the supernatant of Q-Mango, were significantly decreased (*p* < 0.05). In pieces fermentation, the total protein of all the supernatants was significantly increased (*p* < 0.05), while it significantly decreased (*p* < 0.05) in the juices. This result indicated that the effect of fermentation on the distribution of the total protein in the liquid and solid phases of the fermentation system depends on the fruit morphology but not on the type of mango. Fruits are not associated with a high protein content, typically ranging between 0.1 and 1.5%. Despite their low protein content, the proteins in some fruit-based foods are associated with food quality, shelf-life, and health benefits. Traditional fermented food microbes have demonstrated useful proteolytic capabilities as part of their fermentation processes. Multiple reactions, including microbiological, enzymatic, chemical, biochemical, and physical processes, are used in fermentation to convert complex substrates into simple molecules. Enzymes derived from both raw materials and microorganisms are in charge of hydrolysis reactions such as proteolysis in these complex biosystems [47]. Extracellular proteases are produced by some microorganisms and are either attached to the microbial cell membrane or released. These proteases destroy ambient proteins to aid microbial growth, and the process may also aid in food formation [48].

The effect of fermentation on the soluble protein concentration in the supernatant and sediment was measured (Figure 4). There was a greater decrease (*p* < 0.05) in the soluble proteins of supernatants and sediments of juices fermentation and supernatants of fruit pieces fermentation. However, the soluble protein in the supernatants of pieces fermentation was slightly increased. The soluble protein in the supernatant of juices fermentation and all the sediments decreased by 0.07 mg/mL on average while it increased by 0.04 mg/mL in the supernatant of pieces fermentation. The change in the soluble protein was consistent with the changing tendency of the total protein. Moreover, the change in the total protein was greater than that in the soluble protein. Endogenous enzymes and inhibitors are found in a wide variety of raw plant materials, and their activities can have a significant impact on both the degree and specificity of protein hydrolysis during fermentation. Endogenous enzymes, including proteases, are dormant in the native plant matrix; however, a decrease in pH activates some endogenous enzymes [49]. An acidified environment may also activate proteases from both yeast and molds in kombucha. Changes in the pH can also affect plant protein solubility, making proteins more or less accessible for proteolysis during acidifying fermentation. 

The changing tendency of the peptide content in the supernatant and sediment was similar with protein (Figure 5). The reduction in peptides in the supernatant of juices fermentation, sediment of juices fermentation, and sediment of pieces fermentation ranged from 0.07 mg/mL to 0.11 mg/mL, from 0.16 mg/mL to 0.19 mg/mL, and from 0.26 mg/mL to 0.84 mg/mL, respectively. The increase in peptides in the supernatant of pieces fermentation ranged from 0.45 mg/mL to 0.62 mg/mL. In an acidified environment, the deprotonation process of the groups on the polypeptide’s side chain results in peptide chain cleavage and breakage. In a strong acidic environment, the groups in the side chain of polypeptide might be directly hydrolyzed. The peptides were degraded into smaller peptides and amino acids.

Figure 6 shows the change in amino acid nitrogen after fermentation. There was a significant increase (*p* < 0.05) in all samples. The amino acid nitrogen concentration in the supernatant and sediment of juices fermentation gave an increase of 12.83 mg/mL and 23.33 mg/mL, whereas, in pieces fermentation, the supernatant and sediment increased 50.16 mg/mL and 25.67 mg/mL, respectively. During fermentation, the release of nitrogenous substrates depends on the proteases/peptidases produced by the microorganisms [48]. In this study, the total protein, soluble protein, and peptides decreased while amino acid nitrogen increased. Tran et al. reported that yeasts in kombucha can degrade bound amino acids, such as proteins and peptides, into free amino acids [44].

Figure 7 shows the SDS–PAGE profiles of the protein extracts for the supernatant and sediment. No obvious protein bands were observed in all supernatant samples, except that obtained from the juice fermentation of K-Mango. The electrophoretograms showed bands with apparent molecular masses ranging from 25 to 50 kDa in the sediment samples. After fermentation, the concentration and quantity of protein bands in the sample decreased. Particularly, there was an obvious protein band with a molecular weight of approximately 28 KD in the sediments of Q- and K-Mangoes. The molecular weight of the main allergen of mango Man i 1 is 28 KD. The change in SDS-PAGE was consistent with the changing tendency of the proteins and peptides.

Table 1 shows the immunoreactivity of the supernatant and sediment. Most of the sensitization of the samples showed significant differences (*p* < 0.05) after fermentation. The change in the immunogenic reaction with IgE5 was a bit slighter than the other four IgE solutions. The immunoreactivity of all the sediments was reduced, regardless of the fermentation methods. In juices fermentation, the immunoreactivity of both the supernatant and sediment of the three mangoes was reduced. However, the sensitization of the mango supernatant in pieces fermentation was increased. This result was consistent with the change in the protein content, demonstrating that allergenic substances were redistributed in the solid and liquid phases. Hence, fermented mangoes as fruit juices were more effective in reducing the allergenicity. In the present study, the changes in the allergenicity and protein were consistent, indicating that protein is the main allergen of mango fruits. It has been reported that the proteolytic ability of kombucha starters is higher than probiotic starters and yogurt starters [50]. However, in this study, the allergenicity of the fermentation products was not completely removed. Alternatively, fermentation can be combined with other nonthermal techniques to reduce the allergenicity. 

Figure 8 shows a PCA biplot among crude protein; soluble protein; peptides; amino acid; and the IgE reactivity of K-Mango, T-Mango, and Q-Mango before and after fermentation. The cumulative variance contribution rate of PC1 and PC2 was 90.018%. The eigenvalue of the soluble protein and IgE contributed greatly to PC1. From the figure, dots of different mango varieties were gathered together. The closer the distance, the higher the similarity. There was no significant difference between the different mango varieties. The samples of the different fermentation types were distributed in regular and distinct ways. The samples of pieces fermentation gathered together, such as K3, Q3, T3, and they were far away from the dots of K1, Q1, T1, K2, Q2, and T2 (juices fermentation). There was a significant difference between juices fermentation and pieces fermentation. The loadings showed that the crude protein, soluble protein, and peptides had strong positive correlations with the IgE reactivity, while the amino acid had negative correlations with the IgE reactivity. Therefore, the immunoreactivity was closely related to the crude protein, soluble protein, and peptides. The fermentation type had significant effects on the protein degradation and sensitization reduction, while the mango variety had no significant effect. There were no significant interactions between the two factors.

This study showed differences in the protein degradation and sensitization reduction of mangoes between the juices and pieces fermentation. Before juices fermentation, the structure of the mango tissue was mechanically damaged by the blenders. The allergenic proteins in the solid and liquid phases were directly degraded by kombucha during juices fermentation. In juices fermentation, the majority of the proteins, soluble proteins, and peptides in the solution (both in the supernatant and sediment) were degraded into amino acids, resulting in a drop in the proteins, peptides, and allergens in the solid and liquid phases but an increase in amino acids. In pieces fermentation, the transfer channel was opened by microbes, resulting in a rearrangement of the solid and liquid phases. Part of the proteins and peptides in the solid phase were degraded into amino acids, resulting in a drop in the proteins and peptides and a rise in amino acids, while the other part was transported to the liquid phase, leading to an increase of the three substances. The allergenic proteins from the mango block were released to the supernatant, leading to the increased immunoreactivity in the supernatant. Hence, the fermentation with fruit juices was more effective in reducing the allergenicity of mangoes than with fruit pieces. 

## 4. Conclusions

The consumption of mangoes is increasing, but their allergenicity is not conducive to the development of products. In this study, mango juices and pieces were fermented with kombucha. Fermentation can destroy the microstructure of fruit tissue caused by the conversion between insoluble and soluble fractions. In juices fermentation, the structure of the mango tissue was mechanically damaged by the blenders, and the proteins and peptides both in the supernatant and sediment were degraded into amino acids, resulting in a drop in allergens. In pieces fermentation, the transfer channel was opened by microbes, resulting in a rearrangement of the allergenic proteins in the solid and liquid phases, resulting in a drop in allergenicity in fruit pieces but an increase in culture liquid. Protein degradation and sensitization reduction were related to the fermentation type but not mango variety. Hypoallergenic mango juices and blocks can be produced by juice and pieces fermentation, respectively. Fermentation with fruit juices was more effective in reducing the allergenicity of mangoes than with pieces. Thus, fermentation with kombucha is an effective processing method to reduce the sensitization of mangoes.

## Figures and Tables

**Figure 1 foods-12-03465-f001:**
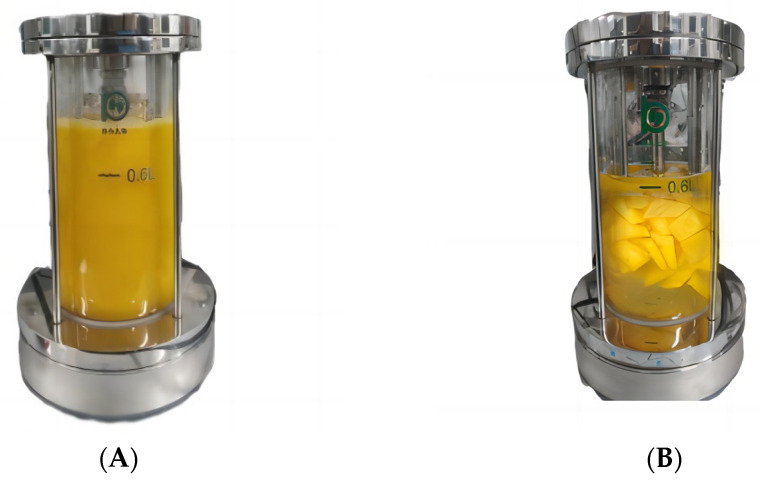
Mango fruits were fermented after beating into juices (**A**) or cutting into pieces (**B**).

**Figure 2 foods-12-03465-f002:**
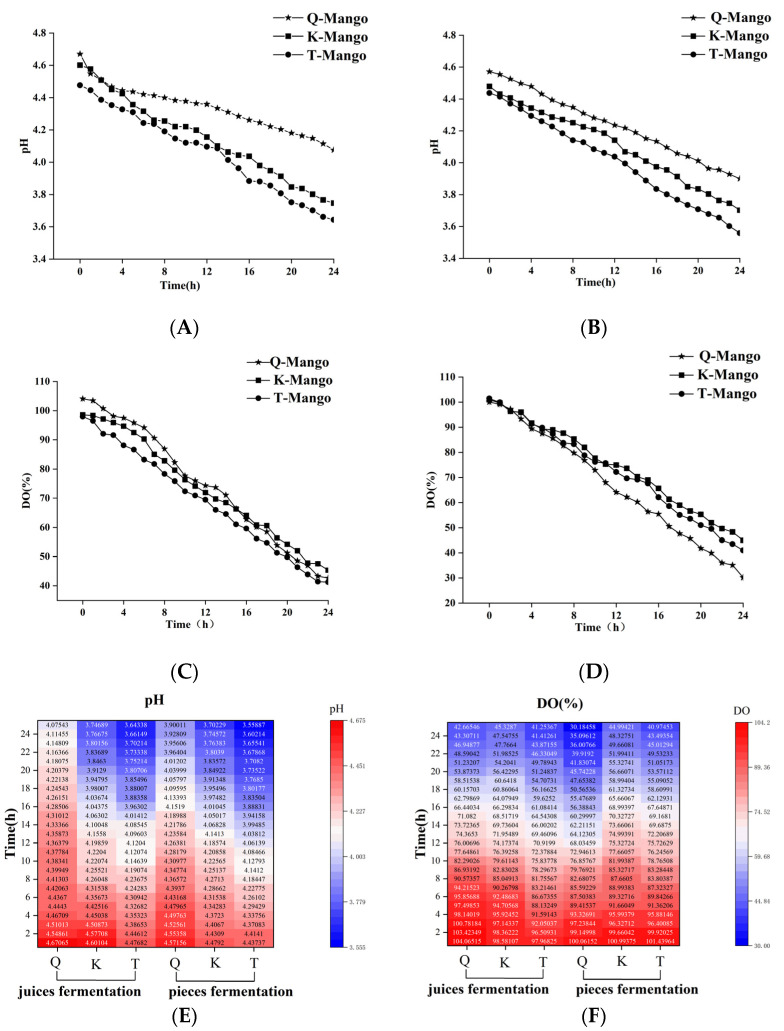
Changes in the pH value and DO concentration of fermentation broth in mango juices fermentation (**A**,**C**) and pieces fermentation (**B**,**D**). The heatmap (**E**,**F**) shows the changing trend in the pH and DO of different types during fermentation. Capital letters Q, K, and T indicate Green Kaew, Keitt, and Tainung mango.

**Figure 3 foods-12-03465-f003:**
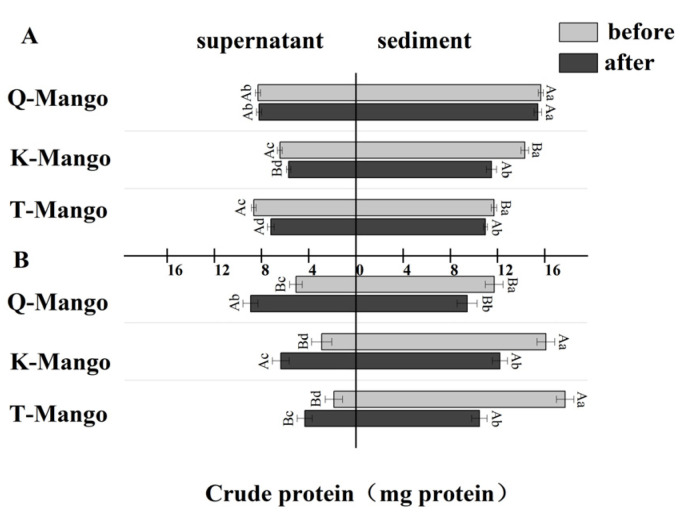
The change in the total protein in the supernatant and sediment of the fermentation broth (**A**) in juices fermentation and (**B**) in pieces fermentation. Capital letters indicate the difference of different fermentation types (*p* < 0.05), and lowercase letters indicate the difference between the same sample before and after fermentation (*p* < 0.05).

**Figure 4 foods-12-03465-f004:**
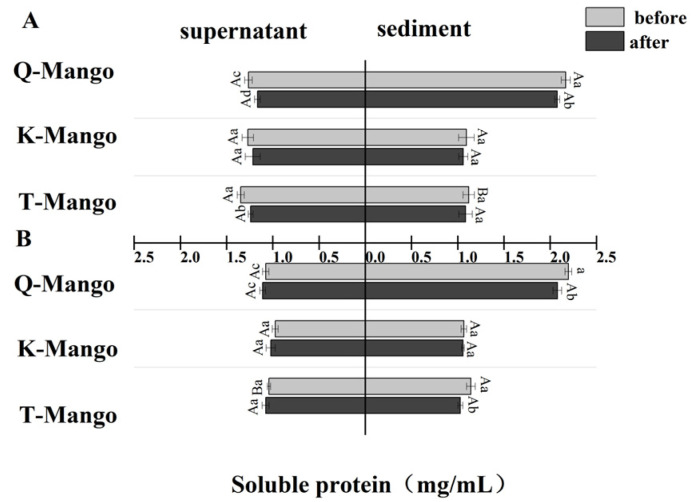
The change in the soluble protein in the supernatant and sediment of the fermentation broth (**A**) in juices fermentation and (**B**) in pieces fermentation. Capital letters indicate the difference of different fermentation types (*p* < 0.05), and lowercase letters indicate the difference between the same sample before and after fermentation (*p* < 0.05).

**Figure 5 foods-12-03465-f005:**
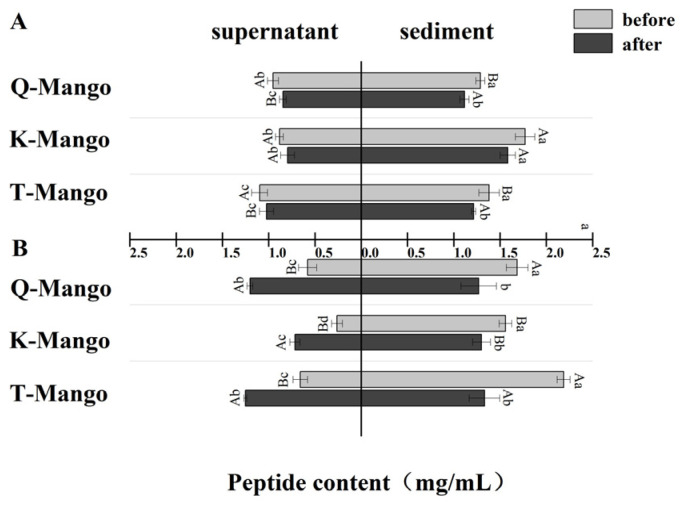
The change in the peptide content in the supernatant and sediment of the fermentation broth (Figure 4) (**A**) in juices fermentation and (**B**) in pieces fermentation. Capital letters indicate the difference of different fermentation types (*p* < 0.05), and lowercase letters indicate the difference between the same sample before and after fermentation (*p* < 0.05).

**Figure 6 foods-12-03465-f006:**
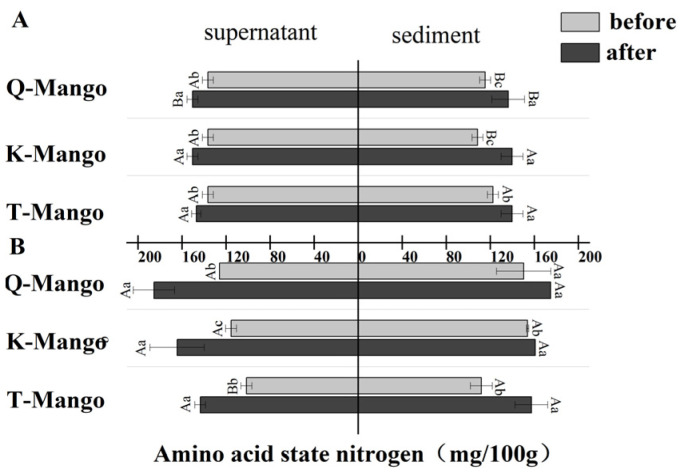
The changes in the amino acid nitrogen content in the supernatant and sediment of the fermentation broth (**A**) in juices fermentation and (**B**) in pieces fermentation. Capital letters indicate the difference of different fermentation types (*p* < 0.05), and lowercase letters indicate the difference between the same sample before and after fermentation (*p* < 0.05).

**Figure 7 foods-12-03465-f007:**
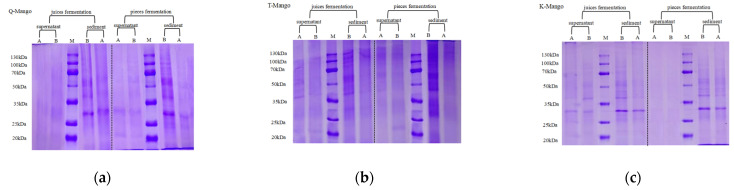
SDS-PAGE pattern of Q-Mango (**a**), T-Mango (**b**), and K-Mango (**c**) before and after fermentation. A = After fermented; B = Before fermented; M = Protein standard; Protein molecular weight markers: 20, 25, 35, 50, 70, 100, and 130 kDa.

**Figure 8 foods-12-03465-f008:**
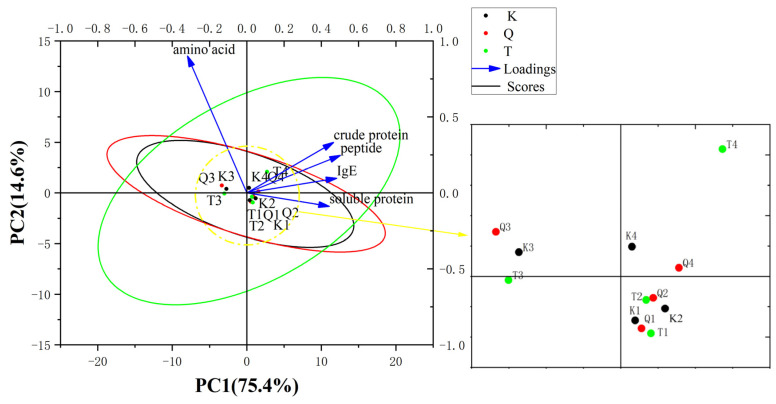
PCA biplots between the crude protein; soluble protein; peptides; amino acid; and IgE reactivity of K-Mango (K), T-Mango (T), and Q-Mango (Q) before and after fermentation. Black, red, and green circles indicate 95% confidence ellipses for K-Mango (K), Q-Mango (Q), and T-Mango (T). K1, Q1, and T1 represent the eigenvalues of the K-Mango, Q-Mango, and T-Mango supernatants obtained from juices fermentation; K2, Q2, and T2 represent the eigenvalues of the K-Mango, Q-Mango, and T-Mango sediments obtained from juices fermentation; K3, Q3, and T3 represent the eigenvalues of the K-Mango, Q-Mango, and T-Mango supernatants obtained from pieces fermentation; and K4, Q4, and T4 represent the eigenvalues of the K-Mango, Q-Mango, and T-Mango sediments obtained from pieces fermentation.

**Table 1 foods-12-03465-t001:** Effect of fermentation on the immune response of human plasma.

IgE (OD_450_)	Mango *	Juices Fermentation				Pieces Fermentation			
		Supernatant		Sediment		Supernatant		Sediment	
		Before	After	Before	After	Before	After	Before	After
IgE1	Q	0.68 ± 0.02 ^Aa^	0.64 ± 0.03 ^Aa^	0.85 ± 0.02 ^Aa^	0.73 ± 0.02 ^Ab^	0.43 ± 0.02 ^Bb^	0.54 ± 0.04 ^Aa^	0.86 ± 0.02 ^Aa^	0.75 ± 0.01 ^Ab^
	K	0.66 ± 0.03 ^Aa^	0.61 ± 0.01 ^Ab^	0.83 ± 0.02 ^Ba^	0.79 ± 0.01 ^Aa^	0.50 ± 0.01 ^Bb^	0.56 ± 0.01 ^Aa^	0.95 ± 0.02 ^Aa^	0.83 ± 0.01 ^Ab^
	T	0.43 ± 0.03 ^Aa^	0.40 ± 0.01 ^Ba^	0.57 ± 0.03 ^Ba^	0.42 ± 0.04 ^Bb^	0.50 ± 0.01 ^Ab^	0.66 ± 0.01 ^Aa^	0.78 ± 0.02 ^Aa^	0.59 ± 0.06 ^Ab^
IgE2	Q	0.22 ± 0.02 ^Aa^	0.16 ± 0.02 ^Bb^	0.46 ± 0.02 ^Aa^	0.36 ± 0.01 ^Ab^	0.30 ± 0.01 ^Ab^	0.56 ± 0.02 ^Aa^	0.42 ± 0.04 ^Aa^	0.31 ± 0.01 ^Ab^
	K	0.26 ± 0.02 ^Aa^	0.23 ± 0.05 ^Aa^	0.46 ± 0.01 ^Aa^	0.28 ± 0.02 ^Ab^	0.27 ± 0.04 ^Aa^	0.31 ± 0.03 ^Aa^	0.38 ± 0.01 ^Aa^	0.35 ± 0.05 ^Aa^
	T	0.43 ± 0.01 ^Aa^	0.36 ± 0.01 ^Ab^	0.51 ± 0.01 ^Ba^	0.39 ± 0.01 ^Bb^	0.23 ± 0.06 ^Bb^	0.40 ± 0.02 ^Aa^	0.65 ± 0.01 ^Aa^	0.58 ± 0.03 ^Ab^
IgE3	Q	0.19 ± 0.02 ^Aa^	0.16 ± 0.01 ^Aa^	0.30 ± 0.01 ^Aa^	0.20 ± 0.01 ^Ab^	0.21 ± 0.06 ^Aa^	0.23 ± 0.09 ^Aa^	0.26 ± 0.07 ^Aa^	0.22 ± 0.04 ^Aa^
	K	0.39 ± 0.05 ^Aa^	0.30 ± 0.06 ^Ab^	0.41 ± 0.1 ^Aa^	0.38 ± 0.1 ^Aa^	0.28 ± 0.06 ^Ab^	0.39 ± 0.1 ^Aa^	0.44 ± 0.2 ^Aa^	0.29 ± 0.08 ^Ab^
	T	0.26 ± 0.03 ^Aa^	0.19 ± 0.02 ^Ab^	0.24 ± 0.02 ^Aa^	0.17 ± 0.01 ^Ab^	0.18 ± 0.01 ^Aa^	0.20 ± 0.04 ^Aa^	0.30 ± 0.1 ^Aa^	0.22 ± 0.04 ^Ab^
IgE4	Q	0.32 ± 0.01 ^Aa^	0.29 ± 0.01 ^Aa^	0.21 ± 0.01 ^Aa^	0.17 ± 0.01 ^Aa^	0.18 ± 0.01 ^Bb^	0.23 ± 0.01 ^Aa^	0.21 ± 0.01 ^Aa^	0.17 ± 0.01 ^Aa^
	K	0.30 ± 0.01 ^Aa^	0.27 ± 0.01 ^Aa^	0.34 ± 0.8 ^Aa^	0.18 ± 0.1 ^Ab^	0.12 ± 0.01 ^Ba^	0.15 ± 0.01 ^Ba^	0.19 ± 0.01 ^Ba^	0.18 ± 0.01 ^Aa^
	T	0.41 ± 0.01 ^Aa^	0.40 ± 0.1 ^Aa^	0.20 ± 0.01 ^Aa^	0.18 ± 0.01 ^Aa^	0.15 ± 0.01 ^Bb^	0.22 ± 0.01 ^Ba^	0.23 ± 0.01 ^Aa^	0.16 ± 0.01 ^Ab^
IgE5	Q	0.26 ± 0.01 ^Aa^	0.23 ± 0.01 ^Aa^	0.27 ± 0.01 ^Aa^	0.26 ± 0.03 ^Aa^	0.24 ± 0.02 ^Aa^	0.26 ± 0.01 ^Aa^	0.34 ± 0.01 ^Aa^	0.29 ± 0.01 ^Ab^
	K	0.24 ± 0.02 ^Aa^	0.21 ± 0.02 ^Aa^	0.28 ± 0.9 ^Aa^	0.24 ± 0.2 ^Aa^	0.24 ± 0.02 ^Aa^	0.24 ± 0.01 ^Aa^	0.27 ± 0.01 ^Aa^	0.25 ± 0.02 ^Aa^
	T	0.26 ± 0.06 ^Aa^	0.20 ± 0.3 ^Ab^	0.23 ± 0.02 ^Aa^	0.20 ± 0.07 ^Aa^	0.25 ± 0.01 ^Aa^	0.26 ± 0.04 ^Aa^	0.3 ± 0.01 ^Aa^	0.29 ± 0.01 ^Ab^

* Capital letters Q, K, and T indicate Green Kaew, Keitt, and Tainung mangoes. Data were represented as the mean ± standard deviation (n = 2 × 3). Capital letters indicate the difference of different fermentation types (*p* < 0.05), and lowercase letters indicate the difference between the same sample before and after fermentation (*p* < 0.05).

## Data Availability

The data presented in this study are available on request from the corresponding author.

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
