# Peer review of "The Differences in Protein Degradation and Sensitization Reduction of Mangoes between Juices and Pieces Fermentation"

_foods, 2023, doi:10.3390/foods12183465_

Round 1

Reviewer 1 Report

The manuscript is an original study. However, it needs significant modifications in order to be fit for publication. 

Line 72: …..“extracted using a sterile cylinder”. This expression must be clarified by adding the brand and model of the cylinder.

Line 159: Statistical analyses must be clarified. Main effects and random effect should be expressed. It is expressed that 3 different mango varieties used in the study. Mango variety is a factor. Second factor is the type of fermentation (juice or piece). Thus, statistical analyzes  should have been carried out on the basis of a factorial trial plan. This is a very important point: Analyzing  just one mango sample three times cannot be considered as a repetition. It is important to review the study from this point of view. In addition, the means of the main effects should be discussed and if there is an interaction between the factors, a figure should be drawn.

According to the findings given in Figure 2  fermentation time (duration) is also a factor. Because of this ,all statistical analyses must be performed from the scratch.  As a result, there are three factors;  A: mango variety, B:fermentation, C: fermentation time

Figure 3:  This figure should be reconsidered taking into account the comments made for Figure 2. If interactions are significant, they must visualized in figures.

Figures 4, 5 and 6: What do “a” and “b” in the figures Express? These should be explained.

Figures should also include Standard deviations.

Line 249 and 285: Literature references must be corrected

Reviewer 2 Report

Manuscript recieved for review analyzes the possibility of mangoes’ kombucha fermentation in juices or pieces form, in order to investigate the changes of proteins and sensitization.  

Title and abstract are adequate.

Intorduction section is elatorate enough, with explanation of researched topic. The aim of the study needs to be more specific regarding obtained results  is detailed.

Material and methodes section is appropriate for the described and conducted testing.

Results section is adequate and sufficient.

It is adviced to combine results and discussion sections into integral one, due to easier tracking of presented research

Conclusion section needs to be more elaborate.  It should be supplemented with more results’ specific conclusionsis.

Some minor corrections are noted in manuscripts’ pdf file.

Decission: major revision

Reviewer 3 Report

The manuscript describes a interesting problem. Requires minor changes:

Abstract: should provide specific information about the research. Please rewrite.

Line 60-64: more should be written about the purpose of the research

Line 83: can the authors explain why 6% sucrose was used?

References to literature are missing in the Material and Methods section (e.g. line 108-111)

Line 164: "During fermentation, the pH value gradually decreases" - it's obvious, please rewrite

Discussion: please refer to the research done

Line 299-300: to be changed, sentence unclear.

Minor editing of English language required

Round 2

Reviewer 1 Report

In my opinion,the manuscript in present form can be accepted for publication.

Reviewer 2 Report

The quality of the manuscript is significantly improved. It is suitable for publication